# Evaluating the Real-World NOx Emission from a China VI Heavy-Duty Diesel Vehicle

**Peng Li \* and Lin Lü**

School of Energy and Power Engineering, Wuhan University of Technology, Wuhan 430063, China; lulinwhut@163.com
\* Correspondence: leepeng@whut.edu.cn; Tel.: +86-134-7708-0396

**Abstract:** The manufacturers of China VI heavy-duty vehicles were required to conduct in-service conformity (ISC) tests by using a portable emissions measurement system (PEMS). The moving averaging window (MAW) method was used to evaluate the NOx emission required by the China VI emission standard. This paper presented the results of four PEMS tests of a China VI (step B) N3 category vehicle. Our analyses revealed that the real NOx emission of the test route was much higher than the result evaluated by the MAW method. We also found the data produced during the urban section of a PEMS test was completely excluded from the evaluation based on the current required boundary conditions. Therefore, in order to ensure the objectivity of the evaluation, this paper proposed three different evaluation methods. Method 1 merely set the power threshold as 10% for valid MAWs; Method 2 reclassified the MAWs into "Urban MAWs", "Rural MAWs" and "Motorway MAWs" according to the vehicle speed. Method 3 reclassified the MAWs into "Hot MAWs" and "Cold MAWs" according to engine coolant temperature. The NOx emission evaluation results for Method 1 were not satisfactory, but those for Method 2 and Method 3 were close to the real NOx emission, the errors were all within ±10%.

**Keywords:** portable emissions measurement system; moving averaging window; real-world NOx emission; heavy-duty diesel vehicle; evaluation method

## 1. Introduction

On-road diesel vehicles produce a great amount of nitrogen oxides (NOx) worldwide, leading to deterioration of the environment and increasing health issues [1,2]. According to updated traffic emission inventory, considering nonlocal trucks, almost 80% of the total vehicular NOx emission in Beijing was emitted by diesel vehicles [2,3].

In the past few decades, increasingly stringent emission regulations had been adopted by many countries (e.g., Europe, US, Korea, Japan) [4,5]. China has also issued "Limits and measurement methods for emissions from diesel fueled heavy-duty vehicles (CHINA VI)" numbered "GB17691-2018" in June 2018.

In order to deal with the increasingly stringent emission standards, the manufacturers generally used selective catalytic reduction (SCR) technology for alleviating NOx emissions at the tailpipe combined with diesel particle filters (DPF) for particulate matter (PM) reduction, diesel oxidation catalysts (DOC) for the oxidation of incomplete combustion products and ammonia slip catalyst (ASC) for the oxidation of $NH_3$ [6,7].

What's more, in the DOC unity, in addition to oxidation of CO and unburned hydrocarbons, NO conversion to $NO_2$ takes place, thus increasing the quite low $NO_2$ concentration in the exhaust gas (about 5% to 10% of total NOx). This increase in $NO_2$ concentration speeds up the passive regeneration process of DPF and, thus, largely affects the decrease in back pressure, enhancing the operating performance and prolonging the life-time of the aftertreatment device [8–10].

Although regulated NOx emission limits had been progressively tightened, many researchers claimed that current diesel vehicles emitted far more NOx under real-world

operating conditions than during laboratory certification testing [1,11,12]. Under this condition, the real driving emissions (RDE) test (also called PEMS test) protocols using a portable emissions measurement system (PEMS)—which is a compact equipment composed by a portable gas analyzer, a global positioning system (GPS) receiver, a data logging system and so on—had already been adopted by many countries (e.g., Europe, US, Korea) to check the on-road conformity of emissions [5,13–17]. In addition, manufacturers were also required to conduct in-service conformity (ISC) testing by using a portable emissions measurement system (PEMS) in China.

The PEMS test regulation establishes the requirements for route composition which must cover a wide range of real-world conditions by accounting with defined shares of urban, rural and motorway operation. Other parameters considered are trip duration, ranges of vehicle speed, cumulative work performed by the engine, etc. [18]. Ambient boundary conditions including altitude and ambient temperature for the PEMS test are also involved in the regulation [14].

However, even with all these requirements, repeatability of PEMS tests is hardly achievable, because the boundary conditions are unique for a PEMS test. For instance, it is scarcely possible to guarantee that the trip time and vehicle speed of two PEMS tests are exactly the same on the same test route, needless to say different test routes [14,19,20].

Therefore, PEMS tests still have some debatable points (e.g., trip composition, boundary conditions and data analysis methods) required further detailed study [14]. For instance, a study performed by Mendoza-Villafuerte et al. [21] revealed that up to 85% of the NOx emissions measured during the tests performed were not taken into consideration if the boundary conditions for data exclusion set in the current legislation were applied.

In order to overcome the repeatability issue of the PEMS tests, many data analysis methods (e.g., the vehicle specific power (VSP) method, the power binning (PB) method, not-to-exceed (NTE) method, moving averaging window (MAW) method, etc.) were introduced for processing the test data [22,23].

Varella et al. [23] tested three different methods (the MAW, PB and VSP), concluding that there were differences between all methods both for $CO_2$ and NOx emissions estimation due to the statistical and numerical treatment from each method. The current data analysis method regulated by the European Community (EC) and China is the moving average window (MAW) method.

This work aims to analyze the data produced during the four PEMS tests. Firstly, to analyze the NOx emission of each section (urban, rural and motorway); then, to calculate the MAW NOx emission under the required boundary conditions; finally, to explore proper methods to evaluate the real-world NOx emission based on MAW method.

## 2. Experiments and Materials

Detailed descriptions of the tested vehicle, test instrumentation and route, MAW method, boundary conditions for a valid PEMS test and data evaluation, judgement rule of pass-fail for emissions are provided in this section.

### 2.1. Tested Vehicle

A heavy-duty diesel vehicle (Figure 1a) which was the type approved to the China VI (step B) standard and registered in August 2019 was used to perform the on-road emissions measurement (PEMS test).

The tested vehicle which covered 2135.3 km at the beginning of Test 1 was equipped with the latest aftertreatment technologies comprised of a diesel oxidation catalyst (DOC) followed by a diesel particulate filter (DPF) in series with a selective catalytic reduction (SCR) catalyst and an ammonia slip catalyst (ASC) in sequence (Figure 1b,c). There are two on-board NOx sensors located at the DOC inlet for the engine output NOx measurement and ASC outlet for tailpipe NOx measurement, respectively (Figure 1c). The main characteristics of the tested vehicle are summarized in Table 1.

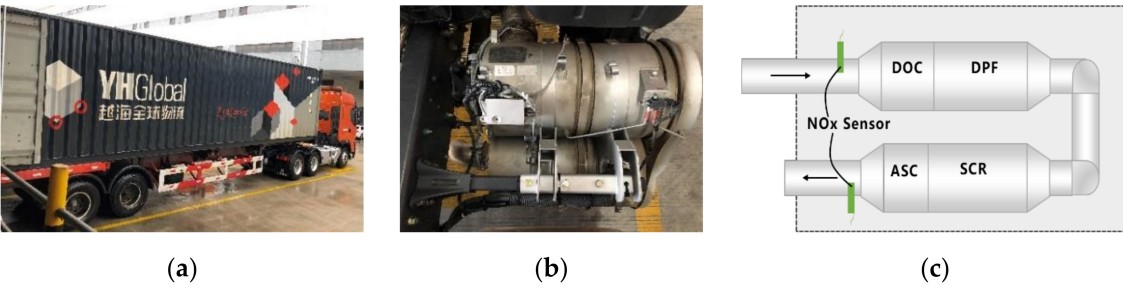

(a) (b) (c)

**Figure 1.** (**a**) Tested vehicle; (**b**) aftertreatment configuration (picture of real products); (**c**) NOx sensors location and aftertreatment configuration (schematic plot).

**Table 1.** Summary of vehicle, engine, aftertreatment, fuel and DEF specifications.

| Type of Engine | XX13600-60 |
|---|---|
| Type of Vehicle | Long–Haul |
| Year of production | 2019 |
| Engine rated power | 441 kW |
| Reference Torque | 3000 Nm |
| WHTC Cycle Work | 38.72 kWh |
| Emission standard | China VI (step B) |
| Aftertreatment System | DOC + DPF + SCR + ASC |
| Gross vehicle weight | kg |
| Payload | 50% |
| Category of vehicle | N3 |
| Fuel | China VI Standard |
| DEF | Adblue (32.5%) |

DEF: diesel exhaust fluid.

### 2.2. Portable Emissions Measurement System (PEMS)

AVL-M.O.V.E-PEMS (Figure 2) consists of tailpipe attachment, heated exhaust lines, exhaust flow meter (EFM), exhaust gas analyzers used to measure concentrations of gaseous emissions (including carbon monoxide (CO), carbon dioxide ($CO_2$), total hydrocarbon (THC), nitrogen monoxide (NO) and nitrogen dioxide ($NO_2$), etc.), PM module, PN module, a global positioning system (GPS) from which we can get vehicle speed, latitude, longitude and altitude, sensors for measuring ambient temperature and humidity, charger, system control, E-box, etc. NOx concentration is calculated by the sum of NO and $NO_2$ concentration. The electrical power needed for the PEMS operation (DC 22~28 V) is supplied by two external batteries.

The PEMS uses flame ionization detection (FID) for THC measurement, non-dispersive infrared (NDIR) for CO and $CO_2$ measurements, non-dispersive ultra-violet (NDUV) for NO, $NO_2$ measurement. The EFM uses a pitot tube based on Bernoulli's principle to calculate mass flow on the basis of airflow differential pressure measurement. The measurement principle and measurement range of gaseous emissions are shown in Table 2.

All emissions are measured on a wet basis, so that no corrections are required for the analysis. The PEMS is warmed up for at least 1.5 h, then zeroed and spanned with calibration gas before the test.

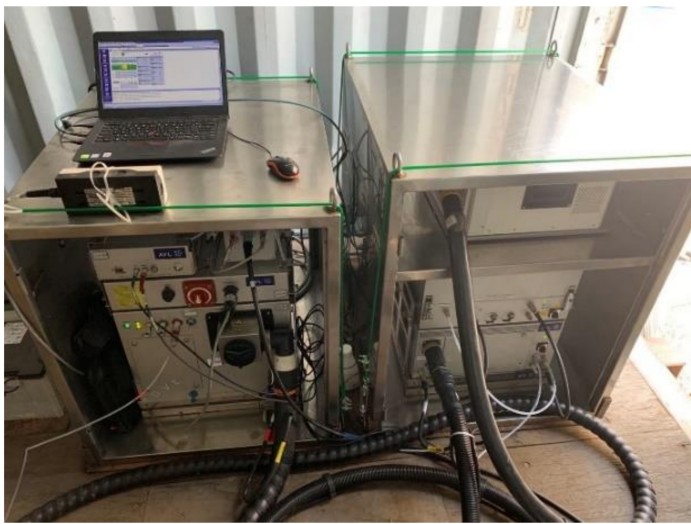

**Figure 2.** Installation of test instrument (AVL-M.O.V.E-PEMS).

**Table 2.** Measurement principle and measurement range of gaseous emissions for the PEMS used.

| Measured Variable | Measurement Principle | Measurement Range |
|---|---|---|
| CO | NDIR | 0~49,999 ppm |
| $CO_2$ | NDIR | 0~20 vol% |
| NO | NDUV | 0~5000 ppm |
| $NO_2$ | NDUV | 0~2500 ppm |
| THC | FID | 0~30,000 ppm |

### 2.3. Test Route

The four PEMS tests were carried out in Suzhou, China along the same route. The test route shall always start with urban driving followed by rural and motorway driving specified in the regulation. We conducted the urban section of the route in the city, the rural and motorway sections on the beltway of Suzhou and a part of the China G2 expressway (Figure 3).

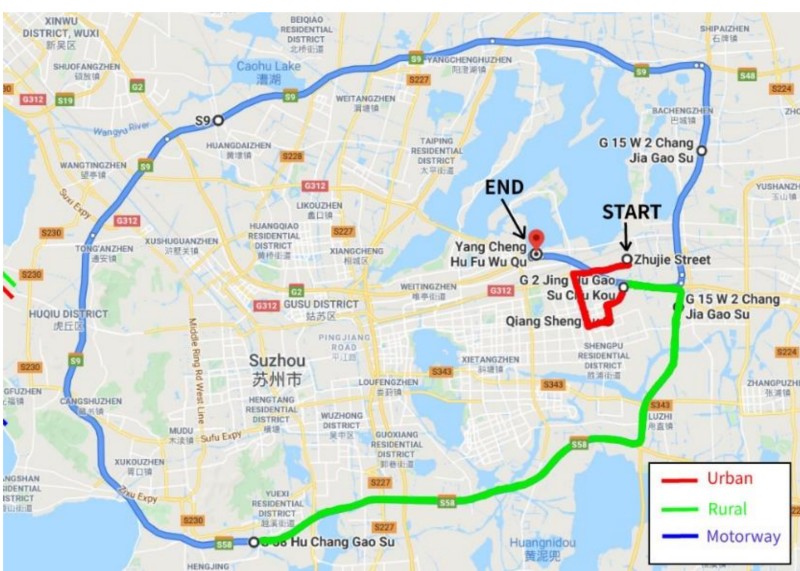

**Figure 3.** Topographic map of the PEMS test route.

For N3 category vehicles, the first short trip (referring to the driving process between the end of one idle speed and the beginning of the next idle speed) with vehicle speed

exceeding 55 km/h is defined as the beginning of the rural section, and the first short trip with vehicle speed exceeding 75 km/h is defined as the beginning of the motorway section. The average vehicle speed of each section shall meet the following requirements: urban section ($\geq$15 to $\leq$30 km/h), rural section ($\geq$45 to $\leq$70 km/h), motorway Section (>70 km/h). The shares of operation shall be expressed as a percentage of the total trip duration, and the trip shall consist of approximately 20% urban, 25% rural and 55% motorway operation. Here, 'approximately' shall mean the target value $\pm$5%.

The entire trip duration is decided by the cumulative work performed by the engine. All the tests considered for the analysis should perform between 4 and 7 times the amount of work performed over the WHTC cycle of the engine.

Moreover, the proportional cumulative positive altitude gain over the entire trip shall be less than 1200 m/100 km, the start and the end point shall not differ in their elevation above sea level by more than 100 m, etc. These requirements were all fulfilled since the test route was performed in a relatively flat area.

### 2.4. Moving Averaging Window (MAW) Method

The emissions shall be evaluated by the MAW method, based on the reference work (the amount of work performed over the WHTC cycle of the engine).

The principle of the calculation is as follows: the mass emissions are not calculated for the complete data set, but for sub-sets of the complete data set, the length of these sub-sets is determined by the work measured over the reference laboratory transient cycle (WHTC for "CHINA VI").

The moving average calculations are conducted with a time increment $\Delta t$ equal to the data sampling period which was set as 500 ms in the four PEMS tests. The end point of the test is taken as the starting point of the first MAW shown in Figure 4.

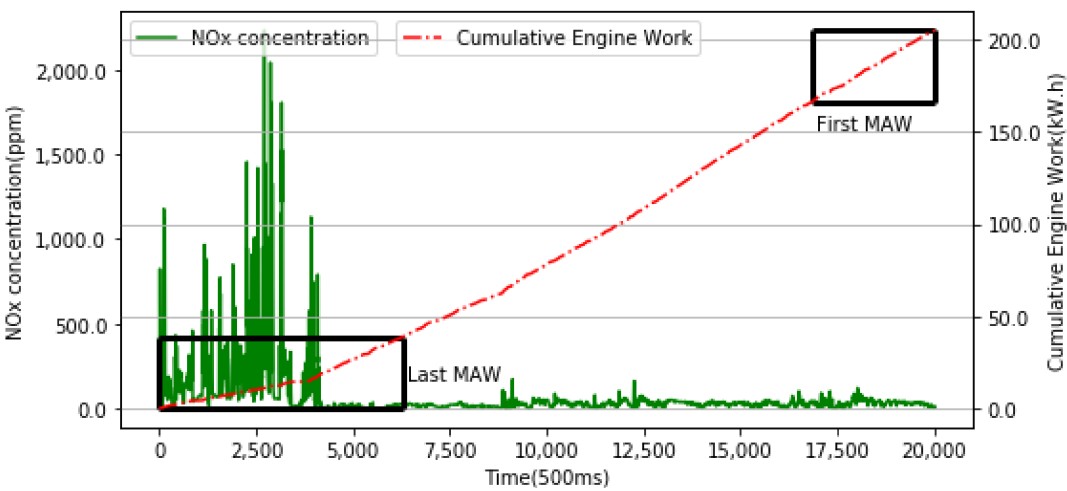

**Figure 4.** Definition of first and last moving averaging windows (MAWs).

The duration $t_{2,i} - t_{1,i}$ of the *i*th averaging window is determined by Equation (1):

$$W(t_{2,i}) - W(t_{1,i}) \geq W_{ref} \tag{1}$$

where:

- $W(t_{j,i})$ is the cumulative engine work measured between the start and time $t_{j,i}$, kWh;
- $W_{ref}$ is the amount of work produced over the WHTC, kWh;
- $t_{2,i}$ shall be selected by Equation (2):

$$W(t_{2,i} - \Delta t) - W(t_{1,i}) < W_{ref} \leq W(t_{2,i}) - W(t_{1,i}) \tag{2}$$

where $\Delta t$ is the data sampling period, equal to 1 s or less.

The brake-specific emissions $EF_p$ (g/kWh) shall be calculated for each window and each pollutant by Equation (3):

$$EF_p = \frac{m}{W(t_{2,i}) - W(t_{1,i})} \tag{3}$$

where:

- $m$ is the cumulative mass of the pollutant of the window, g/window;
- $W(t_{2,i}) - W(t_{1,i})$ is the cumulative engine work during the $i$th averaging window, kWh.

### 2.5. Boundary Conditions for Data Evaluation

The current PEMS procedure for heavy-duty vehicles is defined by a series of boundary conditions that prescribes the amount of data to be taken into consideration for the final analysis including the effectiveness of the test and pass-fail of the pollutants' emissions.

The main boundary conditions for a valid test are as follows:

- Ambient temperature: between −7 and 38 °C.
- Altitude: not more than 2400 m.
- Test route: as abovementioned in "2.3 Test Route" (including trip share and vehicle speed, etc.).
- Cold start: at the beginning of the PEMS test, the engine coolant temperature shall not exceed 30 °C unless the ambient temperature is higher than 30 °C, in this case, the engine coolant temperature shall not be 2 °C higher than the ambient temperature. The data used for emissions evaluation is recorded after the engine coolant temperature has reached 70 °C for the first time or after the coolant temperature is stabilized within ±2 °C over a period of 5 min (whichever comes first but not later than 20 min after engine starts).
- Payload: 10 to 100% of the maximum vehicle payload.
- Cumulative work: 4~7 times the amount of work performed over the WHTC applicable to the engine used by the tested vehicle.
- Selection of valid windows: the valid windows are the windows whose average power exceeds the power threshold of 20% of the maximum engine power. The percentage of valid windows shall be equal or greater than 50%. If the percentage of valid windows is less than 50%, the data evaluation shall be repeated using lower power thresholds. The power threshold shall be reduced in steps of 1% until the percentage of valid windows is equal to or greater than 50%.
- Power Threshold: In any case, the power threshold shall not be lower than 10%, otherwise, the test shall be void.
- The pass-fail conditions for a valid test are as follows:
- The 90th cumulative percentile of the valid windows emissions shall be less than the limit required in the regulation (for China VI, NOx limit: 0.690 g/kWh; CO limit: 6 g/kWh; PN limit: $1.2 \times 10^{12}$ #/kWh);
- NOx concentration is required to be less than or equivalent to 500 ppm for 95% of valid data points.

## 3. Results and Discussion

The data set used for emissions evaluation of the four PEMS tests were all recorded after the engine coolant temperature had reached 70 °C for the first time.

### 3.1. Overall Results

The results of the four PEMS tests conducted on the same route are shown in Table 3. As it can be seen, the four PEMS tests are all valid.

**Table 3.** Results of the four PEMS Tests of the tested vehicle.

|  | Test 1 | Test 2 | Test 3 | Test 4 |
|---|---|---|---|---|
| Altitude not more than 2400 m | Yes | Yes | Yes | Yes |
| Cold start | Yes | Yes | Yes | Yes |
| Average Ambient temperature (°C) | 22.9 | 20.0 | 10.4 | 6.7 |
| Average relative humidity (%) | 96.3 | 69.6 | 66.2 | 65.9 |
| Payload (%) | 50 | 50 | 50 | 50 |
| Urban first (urban-rural-motorway) | Yes | Yes | Yes | Yes |
| Urban share driving (%) | 21.4 | 21.2 | 19.8 | 20.4 |
| Rural share (%) | 22.1 | 27.0 | 25.1 | 25.2 |
| Motorway share (%) | 56.5 | 51.7 | 55.0 | 54.4 |
| Urban driving average speed (km/h) | 25.3 | 24.4 | 21.6 | 23.5 |
| Rural driving average speed (km/h) | 55.4 | 60.9 | 60.9 | 57.7 |
| Motorway driving average speed (km/h) | 75.5 | 73.0 | 75.8 | 76.5 |
| Odometer (km) | 2135.3 | 10,557.4 | 47,443.5 | 82,258.3 |
| Trip distance (km) | 168.8 | 159.1 | 178.4 | 145.6 |
| Trip duration (s) | 9102 | 9117 | 10,009 | 8183 |
| Total Work (kWh) | 195.42 | 179.66 | 205.21 | 175.73 |
| Cumulative work (*$WHTC_{Work}$) | 5.047 | 4.640 | 5.300 | 4.539 |
| Valid MAW Power Threshold (%) | 20 | 17 | 20 | 20 |
| Percentage of valid MAWs (%) | 61.5 | 72.3 | 50.2 | 64.3 |
| 95th NOx concentration (ppm) | 245.1 | 188.0 | 296.2 | 416.2 |
| 90th cumulative percentile of Valid MAWs NOx emission (g/kWh) | 0.068 | 0.551 | 0.438 | 0.337 |
| Test Valid or not | Valid | Valid | Valid | Valid |

Test 1 was conducted on 1 September 2019; Test 2, on 8 October 2019; Test 3, on 26 November 2019; Test 4, on 14 January 2020. The payloads of the four PEMS tests were all 50% of the maximum vehicle load (Gross vehicle weight, 28,800 kg). All of the four PEMS tests started with a cold engine.

The power threshold which shall not be lower than 10% in any case used in the four PEMS tests was 20%, 17%, 20% and 20% respectively, under this circumstance, the percentage of valid MAWs of the four PEMS tests was 61.5%, 72.3%, 50.2%,64.3% respectively. The 90th cumulative percentile of valid MAWs NOx emissions (g/kWh) and 95th cumulative percentile of NOx concentration (ppm) of the four PEMS tests were all within the required limit under the boundary conditions described in Section 2.5 in this paper.

In addition, the odometer at the beginning of Test 1, regarded as customer acceptance testing, was 2135.3 km. The odometer at the beginning of Test 2, regarded as in-service conformity (ISC) testing, was 10,557.4 km. The vehicle odometer shall be at least 10,000 km when carrying out the in-service conformity (ISC) testing. As for Test 3 and Test 4, the odometer at the beginning of the test was chosen to meet the requirements of durability test of the in-service vehicle.

*3.2. Section NOx Emission Analysis*

Before the discussion, it had to be known that the engine output NOx concentration was from the NOx sensor located at the DOC inlet, and the tailpipe NOx concentration was calculated by the sum of NO and $NO_2$ concentration from gas analyzers of the AVL-M.O.V.E-PEMS. In this section, we mainly talked about NOx emission characteristics of each section, especially the urban section.

Figure 5a shows the cumulative mass of engine NOx emission of each section of the test route, and as it can be seen, the least amount of engine NOx emission was emitted during the urban section of the test route because of its shortest test duration. Figure 5b shows the contribution ratio of each section to the total mass of engine NOx emission. Specifically, the contribution ratio of the urban section was 12.39% in Test 1, 10.91% in Test 2, 9.04% in Test 3, 8.74% in Test 4. The contribution ratios of the motorway section were all more than 60% in the four PEMS tests. So, we may conclude that the contribution

of each section to the total mass of engine NOx emission is positively correlated with the trip share.

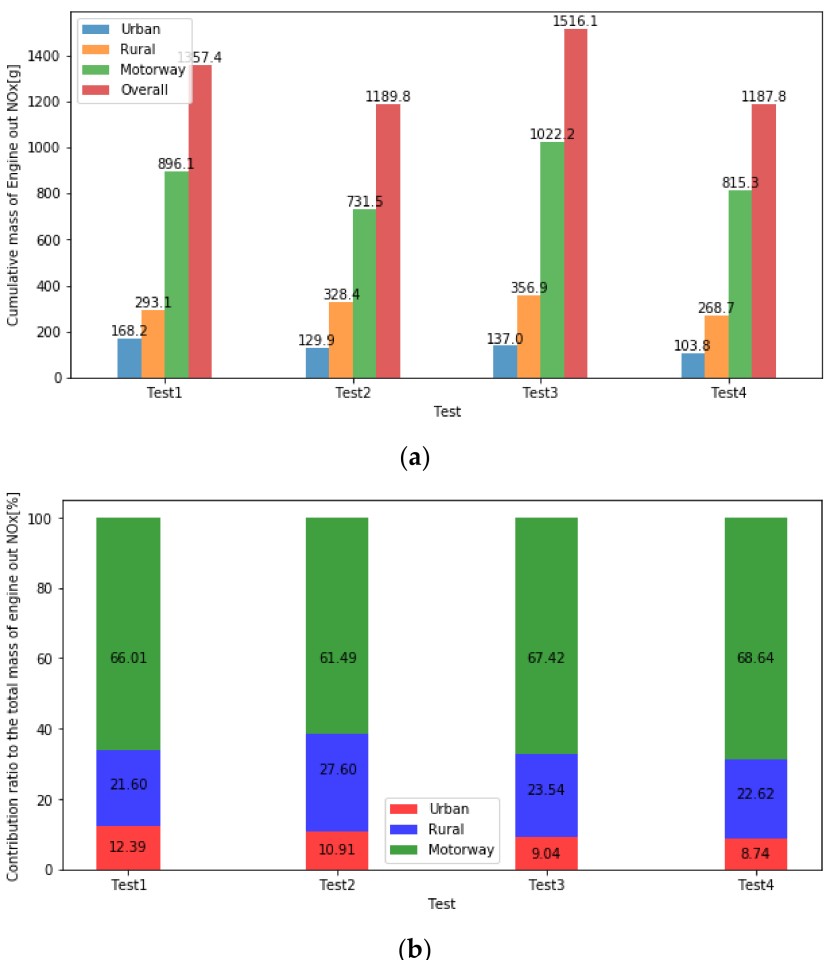

(a)

(b)

**Figure 5.** (**a**) Each section's cumulative mass of engine NOx emission; (**b**) each section's contribution ratio to the total mass of engine NOx emission.

Figure 6a shows the cumulative mass of tailpipe NOx of each section of the test route, as it can be seen, the greatest amount of tailpipe NOx emission was emitted during the urban section in spite of its lowest contribution to the total mass of the engine NOx emission and shortest test duration. The contribution ratio of urban section to the total mass of tailpipe NOx emission was as high as 69.10% in Test 1, 45.25% in Test 2, 55.52% in Test 3, 62.54% in Test 4 (Figure 6b). So, we may conclude that the tailpipe NOx emission of urban section is very terrible for the in-use N3 category heavy-duty vehicles.

Table 4 shows engine output of NOx and tailpipe NOx brake-specific emissions (BSNOx emission: g/kWh) in each section of the test route.

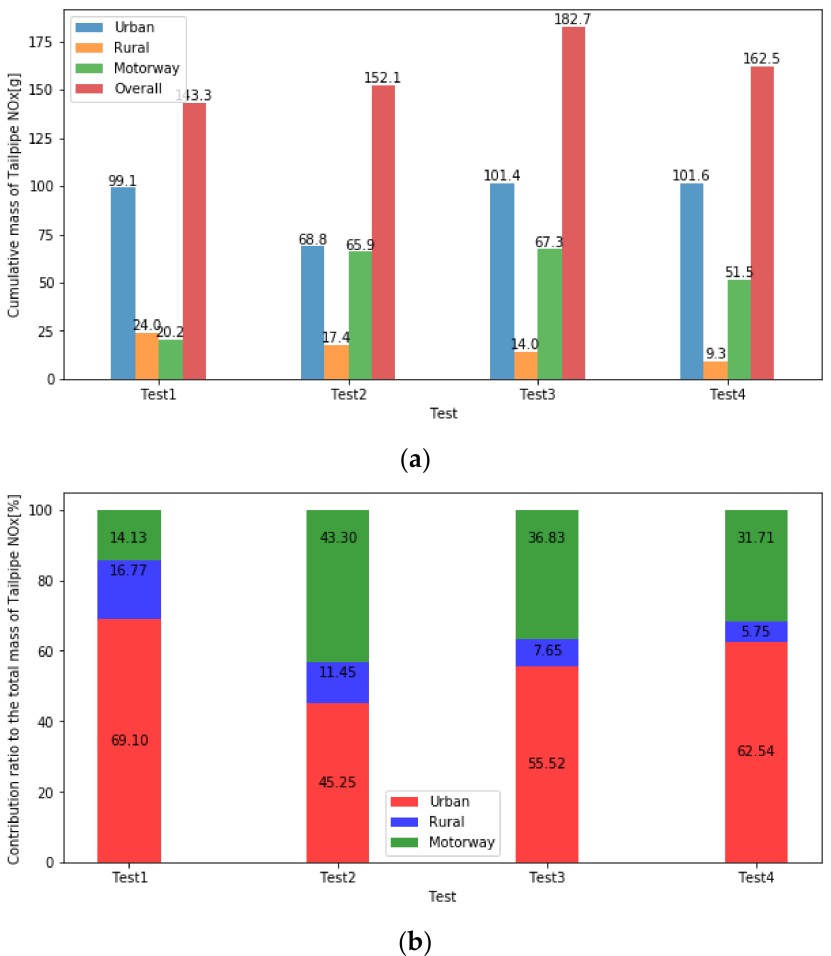

**Figure 6.** (**a**) Each section's cumulative mass of tailpipe NOx emission; (**b**) each section's contribution ratio to the total mass of tailpipe NOx emission.

**Table 4.** NOx brake-specific emissions of each section.

|  | Section | Test 1 | Test 2 | Test 3 | Test 4 |
|---|---|---|---|---|---|
| Engine out NOx emission [g/kWh] | Urban | 8.162 | 6.647 | 8.105 | 7.114 |
|  | Rural | 6.578 | 6.640 | 7.212 | 6.838 |
|  | Motorway | 6.466 | 6.610 | 7.330 | 6.691 |
|  | Overall | 6.665 | 6.622 | 7.363 | 6.759 |
| Tailpipe NOx emission [g/kWh] | Urban | 5.054 | 3.519 | 5.637 | 7.388 |
|  | Rural | 0.395 | 0.370 | 0.344 | 0.271 |
|  | Motorway | 0.059 | 0.502 | 0.413 | 0.317 |
|  | Overall | 0.648 | 0.794 | 0.811 | 0.893 |

As it can be seen, there was a slight difference of the engine out BSNOx emissions between rural section and motorway section in all of the four PEMS tests. The engine output of BSNOx emission of urban section is a slightly higher than that of other sections in Test 1, Test 3 and Test 4, but, in Test 2,the engine out BSNOx emission of each section is almost the same. For the entire trip, the engine out BSNOx emissions of the four PEMS tests were 6.665, 6.622, 7.363 and 6.759 g/kWh, respectively.

The tailpipe BSNOx emissions of rural section and motorway section which were lower than the required limit of NOx emission (0.690 g/kWh) were also lower than that of the entire trip, but, the tailpipe BSNOx emission of urban section was significantly higher than that of other sections. For instance, the tailpipe BSNOx emission of urban section was 8.27 times greater than that of the entire trip in Test 4. For the entire trip, the tailpipe

BSNOx emissions of the four PEMS tests were 0.648 g/kWh, 0.794 g/kWh, 0.811 g/kWh, 0.893 g/kWh, respectively. So, we may conclude that the real-world NOx emissions may get worse as the increase of odometer.

Figure 7 shows the instantaneous emissions of the engine output of NOx, tailpipe NOx and the instantaneous vehicle speed of the entire trip in Test 3. As it can be seen, the tailpipe NOx concentration is almost close to the engine output of NOx concentration in the urban section since catalytic converter requires a certain temperature to work efficiently. A small urea solution injection was registered in the urban section because the temperature of catalytic converter was not high enough. The data points which conducted urea solution injection were only 6.3% of the data points recorded in the whole urban section in Test 3 (Figure 8).

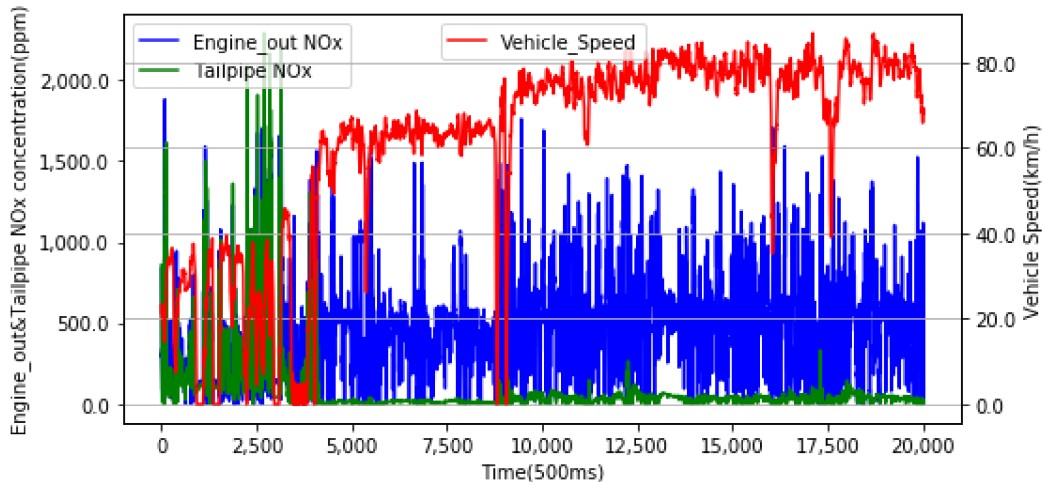

**Figure 7.** The instantaneous emissions of engine output of NOx, tailpipe NOx and vehicle speed of the entire trip in Test 3.

Figure 9a shows the instantaneous engine NOx emission and the vehicle acceleration profile of the entire trip in Test 3. As it can be seen, there was little abrupt acceleration during the rural or motorway section because the vehicle speed was relatively stable during these two sections. Most abrupt positive acceleration occurred during urban section. As shown in Figure 9b, NOx emission peaks were clearly linked to the vehicle acceleration peaks during the urban section, that means abrupt positive vehicle acceleration would lead to worse engine NOx emission. In fact, the urban section of the four PEMS test were mainly conducted on the city road with traffic jam, roundabouts and traffic light, driving under these circumstances may lead to more frequent "stop-go" events where abrupt positive vehicle acceleration would happen.

So, the lower temperature of SCR may lead to a smaller urea solution injection and more frequent "stop-go" events which may lead to higher engine NOx emission together would cause higher tailpipe NOx emission in the urban section.

We see clearly from the above discussion that the urban tailpipe NOx emission plays an important role in the real-world tailpipe NOx emission of N3 category heavy-duty diesel vehicle.

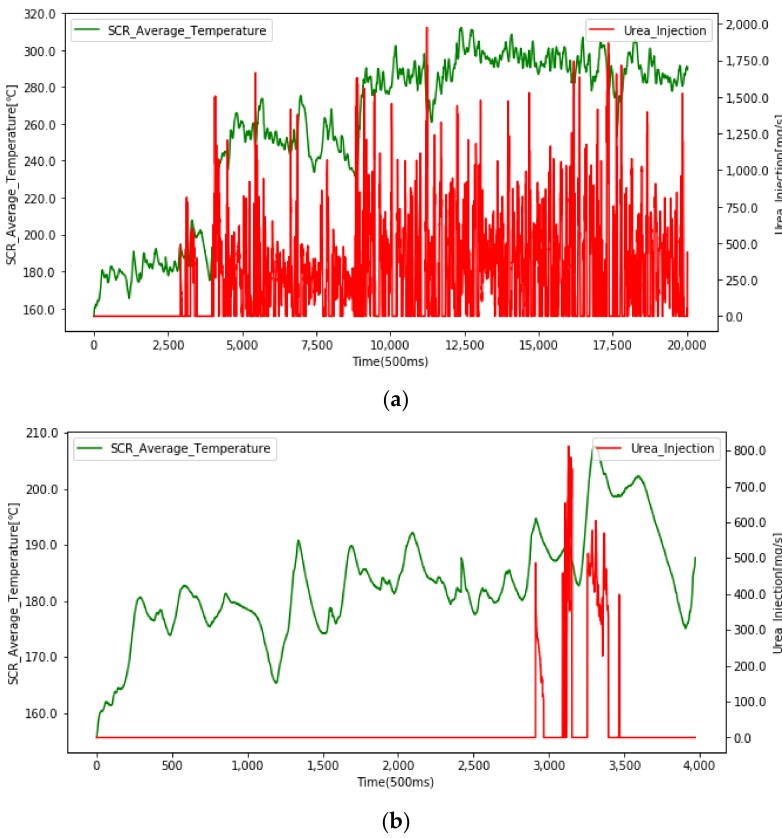

**Figure 8.** The average temperature of selective catalytic reduction (SCR) and the amount of urea solution injection in Test 3; (**a**) for entire trip; (**b**) for urban section.

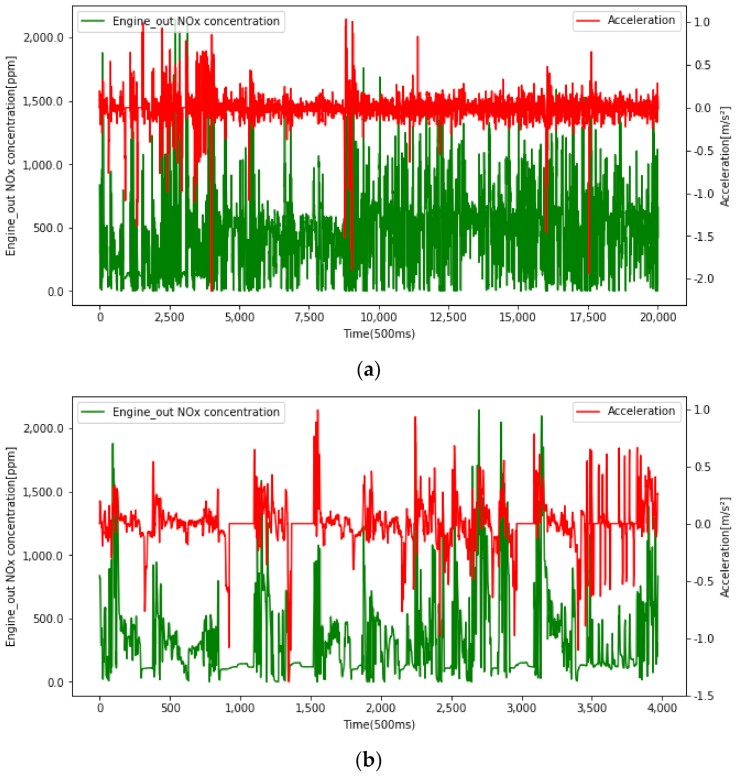

**Figure 9.** The instantaneous engine NOx emission and the vehicle acceleration in Test 3; (**a**) for entire trip; (**b**) for urban section.

*3.3. MAW NOx Emission Analysis*

In this section, the NOx emission we talk about refers to tailpipe NOx emission and all the basic data was from the AVL-M.O.V.E-PEMS. The boundary conditions meet the requirements described in Section 2.5 in this paper.

Table 5 shows the number of MAWs, number of valid MAWs, valid MAWs ratio, power threshold, 90th cumulative percentile of valid MAWs NOx emission (g/kWh), and the real NOx emission of the entire trip (g/kWh), and the difference between entire trip NOx emission and 90th cumulative percentile of valid MAWs NOx emission of the four PEMS test.

**Table 5.** Information of MAW and real NOx emission of the four PEMS tests.

|  | Test 1 | Test 2 | Test 3 | Test 4 |
|---|---|---|---|---|
| MAW Number | 12,291 | 12,513 | 13,742 | 10,432 |
| Valid MAW Number | 7562 | 9047 | 6895 | 6709 |
| Valid MAW Ratio (%) | 61.5 | 72.3 | 50.2 | 64.3 |
| Power Threshold (%) | 20 | 17 | 20 | 20 |
| 90th cumulative percentile of Valid MAWs NOx emission (g/kWh) | 0.068 | 0.551 | 0.438 | 0.337 |
| Real NOx emission of the entire trip (g/kWh) | 0.648 | 0.794 | 0.811 | 0.893 |
| Error 90th Valid MAW to Overall (%) | −89.48 | −30.59 | −45.91 | −62.29 |

The percentage of valid windows was less than 50% when power threshold was 20%, 19%, 18%, respectively in Test 2. So, the final power threshold used to evaluate the NOx emission in Test 2 was 17%.

There was a great difference between the real NOx emission of the entire trip and 90th cumulative percentile of valid MAWs NOx emission (g/kWh). The evaluation results of the four PEMS tests were all lower than the real, the error is −89.48% in Test 1, −30.59% in Test 2, −45.91% in Test 3 and −62.29% in Test 4. Not only that, the evaluated NOx emissions were all lower than the required limit (0.690 g/kWh), but for Test 2, Test 3 and Test 4, the real NOx emissions were higher than 0.690 g/kWh.

Figure 10a shows the distribution, in g/kWh, of the NOx emissions of each MAW versus the average power (% of maximum engine power) of each MAW during Test 3. As it can be seen, the data points in the red rectangle are considered in the final analysis according to the boundary conditions. The higher NOx emissions which are excluded from the final analysis because of the imposed boundary conditions which are mainly concentrated in the MAWs whose average power are higher.

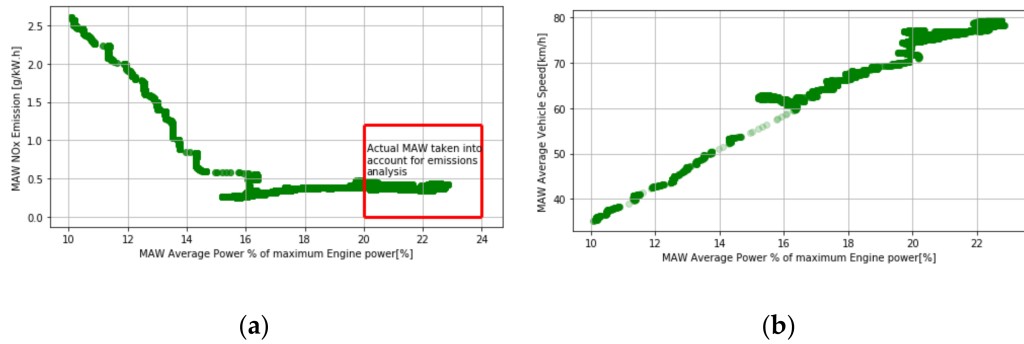

(**a**)                                        (**b**)

**Figure 10.** (**a**) MAWs NOx emissions vs. % of maximum engine power of MAWs during Test 3; (**b**) MAWs vehicle speed vs. % of maximum engine power of MAWs during Test 3.

Figure 10b shows the average vehicle speed and average power (% of maximum engine power) of each MAW. The lower the average power of the MAW is, the lower the

average vehicle speed of the MAW is. As we know, low vehicle speed occurs mainly in the urban section.

Figure 11 shows the distribution of the urban start position, rural start position, motorway start position, test end position, start position of the first valid MAW and end position of the last valid MAW on the timeline. As it can be seen, the end positions of the last valid MAWs are all located in the rural section of the four PEMS tests, that means the valid MAWs obtained by the rules described in Section 2.5 in this paper (in accordance with China VI, GB17691-2018) just represent the emission characteristics of rural and motorway sections.

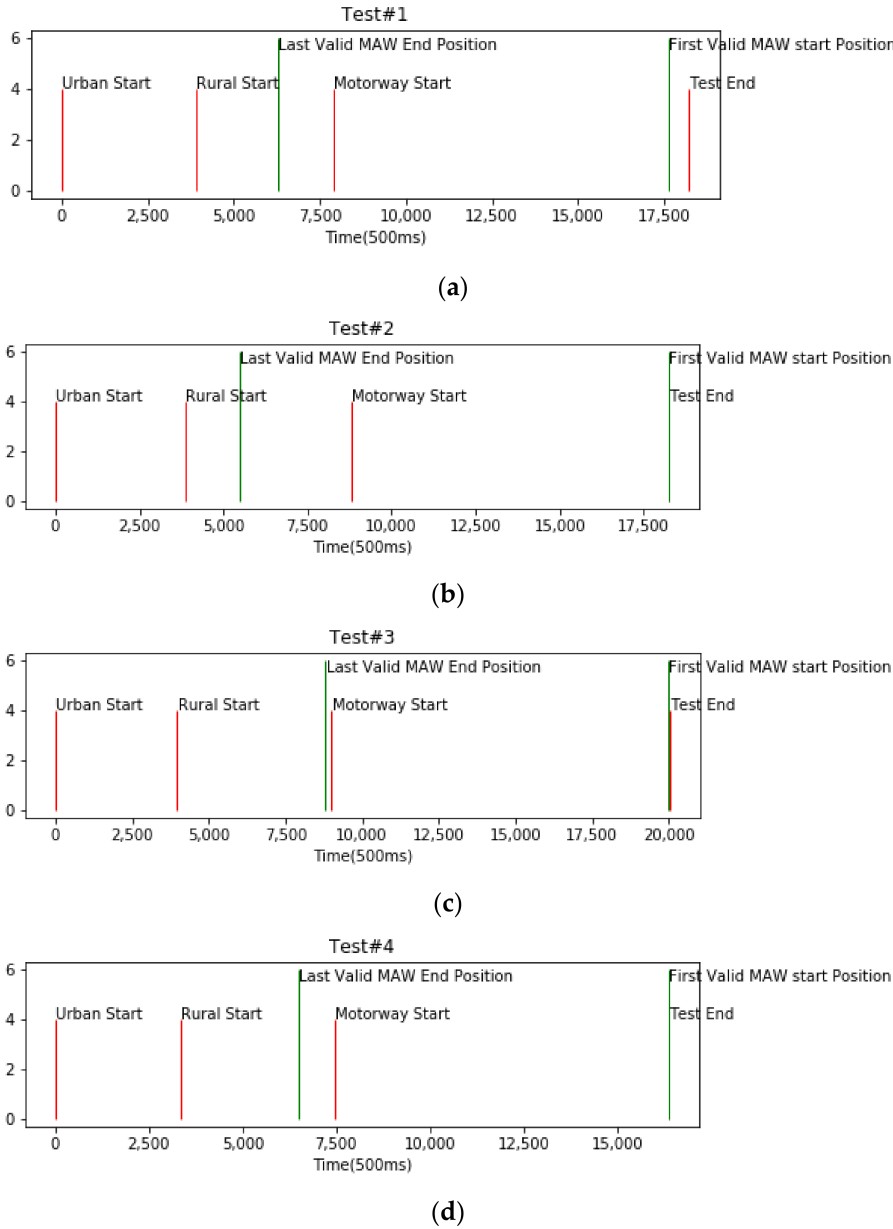

**Figure 11.** Section and first (last) valid MAW distribution information; (**a**) for Test 1; (**b**) for Table 2. (**c**) for Test 3; (**d**) for Test 4.

So, if we still want to evaluate the NOx emission by MAW method, the data during the urban operation must be taken into account for the evaluation and new rules may be needed. Before that, we have to find main influence factors of the MAW NOx emission.

Figure 12 shows a heatmap that reveals the Pearson correlation coefficient between the parameters' mean value or cumulative value of the MAWs in Test 3.

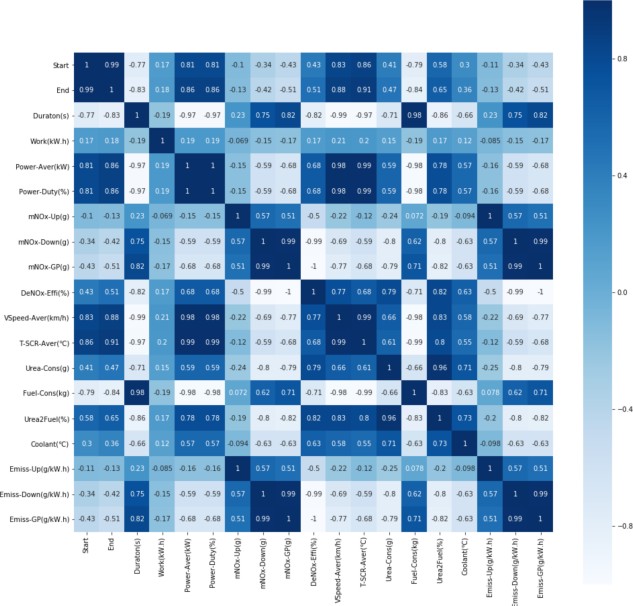

**Figure 12.** Heatmap of MAW parameters inTest#3 (Pearson correlation coefficient).

Table 6 shows the parameters which have a strong correlation to MAW NOx emission according to the heatmap shown in Figure 12. MAW NOx emission is positively correlated with the window duration, the cumulative mass of NOx emission and the cumulative fuel consumption of the MAW; moreover, it is negatively correlated with the average power, NOx conversion efficiency, average vehicle speed, average SCR temperature, urea consumption and urea-fuel ratio of the MAW.

**Table 6.** Pearson correlation coefficient of MAW NOx emission.

|  | Test 1 | Test 2 | Test 3 | Test 4 |
|---|---|---|---|---|
|  | MAW NOx emission [g/kWh] | | | |
| MAW Duration (s) | 0.910 | 0.739 | 0.817 | 0.842 |
| MAW Average Power (kW) | −0.800 | −0.590 | −0.678 | −0.730 |
| MAW Cumulative NOx Mass (g) | 1.000 | 1.000 | 1.000 | 1.000 |
| MAW DeNOx Efficiency (%) | −0.999 | −0.998 | −1.000 | −1.000 |
| MAW Average Vehicle Speed (km/h) | −0.876 | −0.744 | −0.772 | −0.775 |
| MAW Average SCR Temperature (°C) | −0.885 | −0.641 | −0.677 | −0.751 |
| MAW Fuel Consumption (kg) | 0.995 | 0.623 | 0.709 | 0.807 |
| MAW engine coolant(°C) | -0.871 | -0.620 | -0.632 | -0.672 |
| MAW NOx Emission (g/kWh) | 1.000 | 1.000 | 1.000 | 1.000 |

*3.4. The Exploration of Evaluation Methods*

The parameters which have a strong correlation to MAW NOx emission including the MAW duration, the MAW average power, the MAW average vehicle speed, etc. (Table 6) may help us clearly distinguish urban section from rural and motorway sections. Therefore, in order to guarantee the objectivity and accuracy of the evaluation, referring to the baseline already described in Section 2.5 in this paper, three different methods (Method 1~3) were applied to extend the regulatory boundary conditions, details were as follows:

Method 1—in accordance with the baseline except power threshold (PT), and setting the PT as 10% for valid MAWs.

Method 2—in accordance with Method 1 except using the 90th cumulative percentile of the valid MAWs emissions as the result of the PEMS test. Instead, according to the average vehicle speed of the MAWs, redefining the MAWs as "Motorway MAWs" (>70 km/h), "Rural MAWs" (≤70 km/h and ≥reference vehicle speed) and "Urban MAWs" (<reference

vehicle speed), where, the reference vehicle speed is equal to the average vehicle speed of the rural section of the PEMS test. More than that, distributing weighting factors to "Motorway MAWs" (0.55), "Rural MAWs" (0.25), and "Urban MAWs" (0.2) according to the required trip share. The final evaluation results shall be calculated by Equation (4):

$$EF = 90\%ileEF_{Urban\ MAW} * 0.2 + 90\%ileEF_{Rural\ MAW} * 0.25 + 90\%ileEF_{Motorway\ MAW} * 0.55 \tag{4}$$

where: 90%*ile* is 90th cumulative percentile.

Method 3—in accordance with Method 1 except using the 90th cumulative percentile of the valid MAWs emissions as the result of the PEMS test. Instead, redefining the MAWs according to the average engine coolant temperature of the MAWs as "Cold MAWs" (<reference engine coolant temperature) and "Hot MAWs" ($\geq$reference engine coolant temperature), where, the reference engine coolant temperature is equal to the average engine coolant temperature of the entire trip. At the same time, distributing weighting factor to "Cold MAWs" (0.14), "Hot MAWs" (0.86) referring to the WHTC rules. The final evaluation results shall be calculated by Equation (5):

$$EF = 90\%ileEF_{ColdMAW} * 0.14 + 90\%ileEF_{HotMAW} * 0.86 \tag{5}$$

where: 90%*ile* is 90th cumulative percentile.

Due to the uncertainty of the PEMS test procedures, such as trip share, vehicle speed or cumulative work, etc., it is hardly for us to find a constant "Reference Value" to ensure a better universality of Method 2 or Method 3, so, the key point of these two methods were to find a proper "Reference Value". Our data analysis revealed that the average value of a certain section of the test or the entire trip may be a good choice.

Table 7 shows the evaluation results of the Method 1~3, As it can be seen:

**Table 7.** The results of the Method 1~3.

| Method | Parameters | Test 1 | Test 2 | Test 3 | Test 4 |
|---|---|---|---|---|---|
| —— | Real NOx emission of the entire trip (g/kWh) | 0.648 | 0.794 | 0.811 | 0.893 |
| Method 1 | Power Threshold (%) | 10 | 10 | 10 | 10 |
| | NOx emission by Method 1 (g/kWh) | 1.43 | 0.611 | 0.645 | 1.203 |
| | Error1 (Method 1 to Real NOx emission) | 120.7% | −23.1% | −20.4% | 34.6% |
| | MAW Number | 12,291 | 12,513 | 13,742 | 10,432 |
| | Valid MAW Number | 12,291 | 12,513 | 13,742 | 10,432 |
| | Valid MAW Ratio | 100% | 100% | 100% | 100% |
| Method 2 | Power Threshold (%) | 10 | 10 | 10 | 10 |
| | NOx emission by Method 2 (g/kWh) | 0.619 | 0.773 | 0.833 | 0.812 |
| | Error2 (Method 2 to Real NOx emission) | −4.5% | −2.7% | 2.8% | −9.1% |
| | Reference Vehicle Speed (km/h) | 55.4 | 60.9 | 60.9 | 57.7 |
| Method 3 | Power Threshold (%) | 10 | 10 | 10 | 10 |
| | NOx emission by Method 3 (g/kWh) | 0.661 | 0.784 | 0.763 | 0.955 |
| | Error3 (Method 3 to Real NOx emission) | 2.0% | −1.2% | −5.9% | 6.9% |
| | Reference Engine Coolant Temperature (°C) | 78.0 | 80.8 | 80.4 | 80.1 |

For Method 1, when the power threshold (PT) was set as 10%, all of the MAWs of the four PEMS tests were valid, and the data produced during the entire trip was taken into account for the evaluation. Even so, there was also a great difference between the real NOx emission of the entire trip and 90th cumulative percentile of valid MAWs NOx emission (g/kWh), not only that, both positive error and negative error were existed. The error was as high as 120.7% in Test 1, −23.1% in Test 2, −20.4% in Test 3, 34.6% in Test 4. So, we may conclude that just reducing the power threshold (PT) may be not useful enough for an objective evaluation.

For Method 2, the average vehicle speed of rural section of the test was used to distinguish "Urban MAWs" and "Rural MAWs", that because, for N3 category vehicles,

there may be no MAW during the urban operation only. As shown in Figure 13, the cumulative work of urban section of each PEMS test was less than the work performed over the WHTC cycle (38.72 kWh). So, for the four PEMS tests, no MAW was merely composed by the urban operation. The less the cumulative work of urban section, the more the data produced in rural section and used to compose the last MAW.

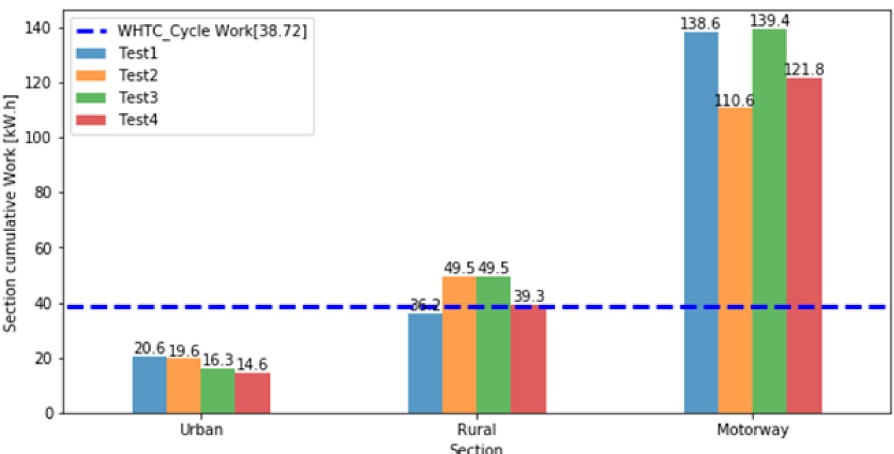

**Figure 13.** Cumulative work of each section of the four PEMS tests.

Figure 14 shows the distribution of the redefined MAWs of the four PEMS tests by Method 2. As it can be seen, the worse polluting windows were categorized as "Urban MAW", most MAWs were categorized as "Motorway MAW" because of its longest test duration and maximum cumulative work. The error between the real NOx emission of the entire trip and the NOx emission evaluated by Method 2 was −4.5% in Test 1, −2.7% in Test 2, 2.8% in Test 3 and −9.1% in Test 4.

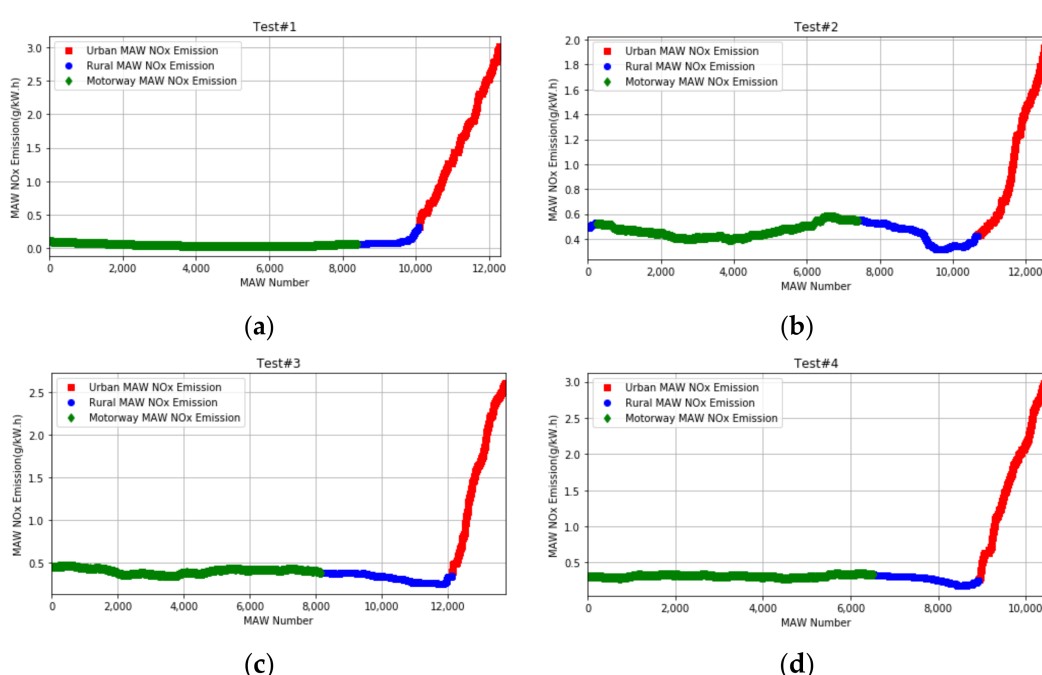

**Figure 14.** Distribution of the redefined MAWs of the four PEMS tests by Method 2; (**a**) for Table 1. (**b**) for Test 2; (**c**) for Test 3; (**d**) for Test 4.

For Method 3, the average engine coolant temperature of the entire trip was used to distinguish "Hot MAWs" and "Cold MAWs". As shown in Figure 15, the engine coolant temperature generally rose gradually and then stabilized around a certain value after

the engine coolant temperature had reached 70 °C for the first time during a PEMS test. Usually, there was no urea solution injection during the "rising period". So, we renamed the "rising period" as "Cold Operation". Via the data analysis, we found that taking the average engine coolant temperature of the entire trip as the "Reference Value" would lead to an exciting result.

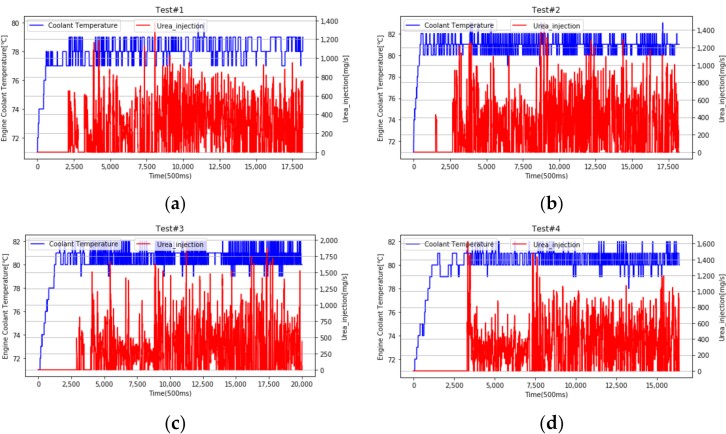

(a)　　　　　　　　　　　　(b)

(c)　　　　　　　　　　　　(d)

**Figure 15.** The instantaneous engine coolant temperature and urea solution injection of the entire trip; (**a**) for Test 1; (**b**) for Test 2; (**c**) for Test 3; (**d**) for Test 4.

Figure 16 shows the distribution of the redefined MAWs of the four PEMS tests by Method 3, only a few of MAWs were regarded as "Cold MAWs" which had worse NOx emission. The error between the real NOx emission of the entire trip and the NOx emission evaluated by Method 3 was 2.0% in Test 1, −1.2% in Test 2, −5.9% in Test 3 and 6.9% in Test 4.

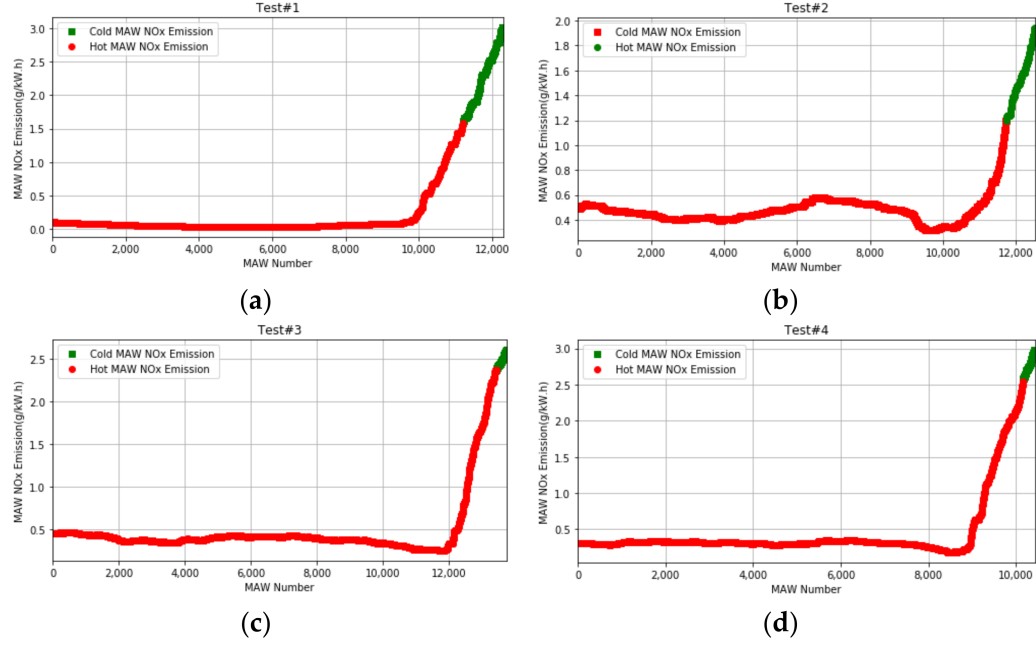

(a)　　　　　　　　　　　　(b)

(c)　　　　　　　　　　　　(d)

**Figure 16.** Distribution of the redefined MAWs of the four PEMS tests by Method 3; (**a**) for Test 1; (**b**) for Test 2; (**c**) for Test 3; (**d**) for Test 4.

To sum it up, the performance of Method 2 or Method 3 was much better than Method 1, and the evaluation errors of Method 2 or Method 3 were all within ±10% (Table 7), that meant vehicle speed and engine coolant temperature could be used as the assistant parameters to classify the MAWs for better heavy-duty vehicle real world NOx emission

evaluation. At the same time, we may conclude that a method that could represent the characteristics of each section (such as "urban section", "cold section") of a PEMS test shall be adopted for the emissions evaluation.

## 4. Conclusions and Outlook

This study had presented and analyzed the data produced by four valid PEMS tests of a China VI (step B), N3 category heavy-duty diesel vehicle. The conclusions of the present study can be summarized to the following:

- The highest tailpipe NOx emissions including the cumulative mass and the brake-specific emission were invariably found during urban operation, which was of great concern for urban air quality and human health. This was mainly because the SCR temperature of the catalytic converter was not high enough to ensure the urea solution injection in most of urban operation. Therefore, heating up SCR rapidly may be an effective means to reduce NOx emission in the urban section.
- For N3 category vehicle, the data produced during the urban section of a PEMS test was excluded from emissions analysis by MAW method due to the higher power threshold (20%) required by boundary conditions. Therefore, a lower power threshold should be used or power threshold boundary should be avoided.
- There was a great difference between the real NOx emission of the entire trip and 90th cumulative percentile of valid MAWs NOx emission(g/kWh)whether the power threshold was set as 10% or 20%.
- The 90th cumulative percentile of valid MAWs NOx emission just represents the emission characteristics of certain sections (rural and motorway for N3 category vehicles) of a PEMS test rather than the entire trip. So, average vehicle speed of the MAWs was used to categorize the MAWs into "Urban MAWs", "Rural MAWs" and "Motorway MAWs"; average engine coolant temperature of the MAWs was used to categorize the MAWs into "Hot MAWs" and "Cold MAWs". The evaluation results of the NOx emission of these two kinds of categorized MAWs were close to the real NOx emission, and the errors were all within ±10%.

In this work, we had pointed out the insufficient of the current evaluation method for heavy-duty vehicle real world NOx emission. Future studies should focus on the following aspects:

- The control and evaluation for NOx emission of cold start (engine coolant temperature less than 70 °C) or low load operation conditions.
- More individualized boundary conditions or MAW rules for the real-world NOx emission of each category vehicle, such as the definition of power thresholds (PT), valid MAW, weights factors, etc.
- The real amount (g, g/kW.h or g/km) of heavy-duty diesel vehicles' real world NOx emission, especially the urban section.

**Author Contributions:** Conceptualization, P.L. and L.L.; methodology, P.L. and L.L.; formal analysis, P.L.; investigation, P.L. and L.L.; resources, L.L.; data curation, P.L.; writing—original draft preparation, P.L.; writing—review and editing, L.L.; visualization, P.L.; supervision, L.L.; project administration, L.L.; funding acquisition, L.L. All authors have read and agreed to the published version of the manuscript.

**Funding:** This research was funded by National Natural Science Foundation of China (NSFC), grant number 51679176.

**Institutional Review Board Statement:** Not applicable.

**Informed Consent Statement:** Not applicable.

**Data Availability Statement:** Data sharing not applicable.

**Acknowledgments:** The authors are thankful to all the personnel who either provided technical.

**Conflicts of Interest:** The authors declare no conflict of interest.

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
