# Peer review of "Evaluating the Real-World NOx Emission from a China VI Heavy-Duty Diesel Vehicle"

_applsci, doi:10.3390/app11031335_

Round 1

Reviewer 1 Report

An interesting topic addressed by the authors.

Moderate English changes required and spelling (especially for the chemical annotations). 

Section 2 can be improved by adding a short introduction that describes the concept and continuing with the subsections. 

Figure captions can be more detailed in some cases (see fig 2, fig 3 ).

Author Response

  1. Moderate English changes required and spelling (especially for the chemical annotations).

[Answer]: Make changes as suggestions.

[Revision]: The spelling and chemical annotations of this paper has been carefully checked and revised. As for chemical annotations, NH3, CO2, NO2 were all changed to NH3, CO2, NO2 respectively.

  1. Section 2 can be improved by adding a short introduction that describes the concept and continuing with the subsections. 

[Answer]: Make changes as suggestions, a short introduction had been added. (See line 86~88)

[Revision]: Detailed description of the tested vehicle, test instrumentation and route, MAW method, boundary conditions for a valid PEMS test and data evaluation, judgement rule of pass-fail for emissions had been provided in this chapter.

  1. Figure captions can be more detailed in some cases (see fig 2, fig 3 ).

[Answer]: Make changes as suggestions. Figure captions of Fig.2 and Fig.3 had been detailed (See line115, line 148)

[Revision]: Figure 2. Installation of test instrument (AVL-M.O.V.E-PEMS); Figure 3. Topographic map of the PEMS test route.

Reviewer 2 Report

The manuscript presents very actual and interesting work results regarding heavy-duty vehicle real world emissions of NOx. This item is under active discussion at UN ECE level because of international harmonized test procedure development, so the presented work is important contribution to that process.

The manuscript good enough describes the aim, test object, test procedure and results with deep discussion and comprehensive conclusions.

In the text there are some mistakes or inaccuracies:

  1. 3. table. 1 and p.6, string 204 Gross vehicle weight 288 t seems to be very high.
  2. 3 abbreviation “DEF” is not described.
  3. 7. table 3 and p.12, table 5, test 1 90th cumulative percentile of Valid MAWs NOx emission[g/kWh] = 0.068 is very low.
  4. 9. table 4, test 1 section Motorway Tailpipe NOx emission [g/kWh] = 0.059 is very low.

Please correct or explain these values.

Author Response

  1. 3 ,table. 1 and p.6, string 204 Gross vehicle weight 288 t seems to be very high.

[Answer]: We had made a clerical error, the Gross vehicle weight is 28.8t.

[Revision]: The value in Table1, and Line 216 of Revised Manuscript had been changed to 28800kg.

  1. 3 abbreviation “DEF” is not described.

[Answer]: Make changes as suggestions, See line 103

[Revision]: DEF: Diesel Exhaust Fluid

  1. 7 table 3 and p.12, table 5, test 1 90th cumulative percentile of Valid MAWs NOx emission[g/kWh] = 0.068 is very low

[Answer]:The result of the Test#1 was lower than the other tests, because, the tailpipe NOx was very low (for most sampling point, not exceed 10ppm)during the motorway section of Test #1, as described in this paper, the data for valid MAWs was almost from the motorway section. So, 90th cumulative percentile of valid MAWs NOx emission [g/kWh] = 0.068 is very low, but it is the real. Perhaps during the run-in period (Odometer for Test#1was 2135.3km), the emissions of the tested vehicle were not stable.

[Revision]: No revision has been made.

  1. 9 table 4, test 1 section Motorway Tailpipe NOx emission [g/kWh] = 0.059 is very low.

[Answer]: The result of the Test#1 was lower than the other tests, because, the tailpipe NOx was very low (for most sampling point, not exceed 10ppm) during the motorway section of Test #1. Perhaps during the run-in period (Odometer for Test#1was 2135.3km), the emissions of the tested vehicle were not stable.

[Revision]: No revision has been made.

Reviewer 3 Report

This is an interesting work addressing an important topic. It has the potential to be published in Applied Sciences. However, I have the following comments that the authors have to implement in the revised manuscript before publication.

1) In order to give a more complete picture, the authors should highlight that, in the DOC unity, in addition to oxidation of CO and unburned hydrocarbons, NO conversion to NO2 takes place, thus increasing the quite low NO2 concentration in the exhaust gas (about 5% to 10% of total NOx). NO2 oxidizes soot more effectively than O2 at lower temperature (about 300 °C lower). This increase in NO2 concentration speeds up the regeneration process of the DPF and, thus, largely affects the decrease in back pressure, enhancing the operating performance and prolonging the life-time of the device. This issue is well explained in the following paper that should be cited: Catalysts, 2020, 10(11), 1307 (doi: 10.3390/catal10111307).

2) Discussion/Conclusions - In the discussion, the authors have to highlight better the practical impact of the results obtained in this work. This should also be done at the end of the section "Conclusions" - conclusions cannot be a list of results.

3) Conclusions - The authors should also give an outlook on future research work. What does this work pave the way for? The authors should comment on this issue.

4) The English language should be improved throughout the manuscript.

I'm willing to review the revised manuscript.

Author Response

  1. In order to give a more complete picture, the authors should highlight that, in the DOC unity, in addition to oxidation of CO and unburned hydrocarbons, NO conversion to NO2 takes place, thus increasing the quite low NO2 concentration in the exhaust gas (about 5% to 10% of total NOx). NO2 oxidizes soot more effectively than O2 at lower temperature (about 300 °C lower). This increase in NO2 concentration speeds up the regeneration process of the DPF and, thus, largely affects the decrease in back pressure, enhancing the operating performance and prolonging the life-time of the device. This issue is well explained in the following paper that should be cited: Catalysts, 2020, 10(11), 1307 (doi: 10.3390/catal10111307).

[Answer]: Make changes as suggestions. (See line39~44 and line 531~536)

[Revision]:What’s more, in the DOC unity, in addition to oxidation of CO and unburned hydrocarbons, NO conversion to NO2 takes place, thus increasing the quite low NO2 concentration in the exhaust gas (about 5% to 10% of total NOx).This increase in NO2 concentration speeds up the passive regeneration process of DPF and, thus, largely affects the decrease in back pressure, enhancing the operating performance and prolonging the life-time of the aftertreatment device. [8,9,10]

  1. Lisi, L., G. Landi, and V. Di Sarli, The Issue of Soot-Catalyst Contact in Regeneration of Catalytic Diesel Particulate Filters: A Critical Review. Catalysts, 2020. 10,1307.
  2. Jiaqiang,E.; Xie,L.; Zuo,Q.; Zhang,G. Effect analysis on regeneration speed of continuous regeneration-diesel particulate filter based on NO2-assisted regeneration. Atmos. Pollut. Res. 2016, 7, 9–17.
  3. Jiaqiang, E.; Zuo, W.; Gao, J.; Peng, Q.; Zhang, Z.; Hieu, P.M. Effect analysis on pressure drop of the continuous regeneration-diesel particulate filter based on NO2 assisted regeneration. Appl. Therm. Eng. 2016 ,100, 356–366.

  1. Discussion/Conclusions - In the discussion, the authors have to highlight better the practical impact of the results obtained in this work. This should also be done at the end of the section "Conclusions" - conclusions cannot be a list of results. 

[Answer]: Make changes as suggestions. (See line 465~471)

[Revision]: To sum it up, the performance of Method #2 or Method #3 was much better than Method #1, and the evaluation errors of Method #2 or Method #3 were all within ±10%(Table 7), that meant vehicle speed and engine coolant temperature could be used as the assistant parameters to classify the MAWs for better heavy-duty vehicle real world NOx emission evaluation. . As for the universality of Method #2 and Method #3, more test data would be needed for further validation. At the same time, we may conclude that a method that could represent the characteristics of each section (such as “urban section”, “cold section”) of a PEMS test shall be adopted for the emissions evaluation.

  1. Conclusions - The authors should also give an outlook on future research work. What does this work pave the way for? The authors should comment on this issue.

[Answer]: Make changes as suggestions. (See line 497~506)

[Revision]: In this work, we had pointed out the insufficient of the current evaluation method for heavy-duty vehicle real world NOx emission. Future studies should focus on the following aspects:

  • The control and evaluation for NOx emission of cold start (engine coolant tem-perature less than 70℃) or low load operation conditions.
  • More individualized boundary conditions or MAW rules for the real world NOx emission of each category vehicle, such as the definition of power thresholds (PT), valid MAW, weights factors, etc.
  • The real amount (g, g/kW.h or g/km) of heavy-duty diesel vehicles’ real world NOx emission, especially the urban section.

  1. The English language should be improved throughout the manuscript.

[Answer]: Make changes as suggestions.

[Revision]: The language of this paper has been carefully checked and revised.

Round 2

Reviewer 3 Report

The authors have addressed my comments in a satisfactory manner. Overall, the manuscript has been improved after revision. Therefore, it can be accepted for publication in Catalysts.

This manuscript is a resubmission of an earlier submission. The following is a list of the peer review reports and author responses from that submission.